# Extending Shelf Life of Cabbage (*Brassica oleracea* cv. K-Y Cross) by Using Vacuum Precooling and Modified Atmosphere Packaging

Jung-Yu Liu [1], Che-Wei Liu [1], Wei-Ling Chen [2], Min-Chi Hsu [3], Huey-Ling Lin [1] and Chang-Lin Chen [1,*]

[1] Department of Horticulture, National Chung Hsing University, 145 Xingda Road, Taichung City 40227, Taiwan; lzw0217@gmail.com (C.-W.L.)

[2] Taichung Agricultural Research and Extension Station, Ministry of Agriculture, Changhua 51544, Taiwan

[3] Taiwan Agricultural Research Institute, Ministry of Agriculture, 189 Zhongzheng Rd., Wufeng Dist., Taichung City 413008, Taiwan

*   Correspondence: changlinchen@dragon.nchu.edu.tw; Tel.: +886-4-22840340 (ext. 307)

**Abstract:** The cabbage cultivar 'K-Y cross' is the most important vegetable crop in Taiwan, but it is not easy to produce through the whole year because of the hot summer. For year-round supply, the different postharvest methods such as precooling and modified atmosphere packaging (MAP) were applied to extend the storage period. The respiration rate and ethylene production were measured in a wide range of temperatures from 1 to 30 °C, and the end of precooling temperature was recommended to be around 7 °C (3/4 precooling times). Vacuum precooling (VC) took cabbages only 30 min to decrease from around 27 °C to 7 °C, which was 19 times faster than room precooling (RC). After 3/4 precooling times, temperature distribution from surface to central parts of cabbages was shown to be more uniform by VC than that by RC. After VC, the MAP was applied to maintain the quality traits of cabbages. The passive MAP could decrease the weight loss, yellowing, and physiological disorders and maintain around 2% $O_2$ and 4% $CO_2$ in the 0.03 mm LDPE bags at 1 °C after 12-week storage. Therefore, the quality of cabbages could be maintained over 3 months by combing VC and passive MAP.

**Keywords:** precooling time; vacuum cooling; room cooling; ethylene production; respiration rate

## 1. Introduction

Cabbages (*Brassica oleraceae* var. *capitata*.) are economically important crops, and there are many different types over the world. According to the growing period in Taiwan, there are two different types (early and late maturity) of cabbages. Normally, the storage life of the early maturity varieties is less than 6 weeks, but the late ones can stand for 24 weeks [1]. Even if there are many different cultivars in Taiwan, the most commonly cultivated cabbage, 'K-Y cross', is the early maturity variety, which holds 80% market share because of the sweetness and texture [2]. This cultivar is grown in the highlands in summer and in lowlands during fall and winter in Taiwan. Because the climate in Taiwan is wet and hot in later spring and in summer, the best period for growing cabbages is between fall and winter when they are harvested the next spring. Therefore, there is a shortage in later spring to summer because of its short storage life. If cabbages grown in winter (harvest in spring) can be stored until summer, a supply of year-round cabbages from local farmer is possible.

Precooling in the cold chain is essential to improve storage ability by rapidly removing heat from commodity. This process extends the storage period because low temperature not only reduces the respiration rate and ethylene production of agricultural products but also inhibits the growth of microbes that may cause spoilage. There are many different precooling methods, including room precooling (RC), hydro precooling, forced-air precooling, liquid ice precooling, and vacuum precooling (VC) [3]. Currently, cabbages in Taiwan

are mainly precooled though RC. Although RC is the easiest method, it takes the longest precooling time. In contrast, the VC is the fastest method because of a rapid evaporation of water from the surface of products. Nevertheless, the major disadvantage of VC is the loss in weight of 1% for every 11 °C reduction in temperature [3]. Because of the high efficiency of removing field heat, it has often been used for precooling fruits, vegetables, or cut flowers. For iceberg lettuces, VC is about 13 times faster than RC, and the temperature drops at the surface and at the center are very similar because of the structure [4].

The postharvest vegetable quality decreases because of water loss, yellowing, decay, and physiological disorder. Therefore, the storage condition should maintain low temperature and high and stable humidity to avoid the damages in long-term storage. On the other hand, the major physiological storage disorder, including black spot and grey speck, can be reduced by a controlled atmosphere (CA) with 2–3% $O_2$ and 5–6% $CO_2$ [5–7]. Although low concentrations of $O_2$ and high levels of $CO_2$ have proved to be effective in delaying yellowing and in reducing the incidence of decay in Chinese white cabbages [6] and white cabbages [8], the equipment of CA is costly. Alternatively, modified atmosphere packaging (MAP) is a widely applied technology for maintaining quality during storage at low temperature [9]. The agricultural products packaged in polyethylene (PE) bags consistently consume $O_2$ and release $CO_2$, so the atmosphere composition will result in high $CO_2$ concentrations and low $O_2$ concentrations after balance. This atmosphere not only slows down the respiration rate, ethylene production, and water loss but also reduces the enzyme activities and physiological disorder [10–13].

The most commonly cultivated 'K-Y cross' cabbages with sweetness and crispness have low potential for storage in Taiwan. Therefore, VC and MAP was applied to extend the storage and shelf life to fit the demand between the later spring and summer. The objectives of this investigation were to (1) compare the difference of cabbages by VC and RC, (2) optimize precooling parameters, (3) combine VC and MAP to extend the shelf life, and (4) survey the change of quality after storage and shelf.

## 2. Materials and Methods

### 2.1. Respiration Rate and Ethylene Concentration at Different Temperatures

The 'K-Y cross' cabbages cultivated in Changhua county of Taiwan (23.848672, 120.422647) were harvested from January to April 2021 and in December 2022. The weight of cabbages of around 2 Kg was used in this experiment and transported to the packaging site for precooling. After the cabbages were stored in incubators or coolers with temperatures from 1 to 30 °C overnight, three cabbages from different temperatures were weighed and closed in the three individual 6.6 L tanks for one hour. A 1 mL syringe sucked the sample from the tanks to measure the respiration by an infrared $CO_2$ analyzer (X-stream enhanced XEGK, Rosemount, Shakopee, MN, USA). The respiration rate was expressed as mL $CO_2$ $Kg^{-1}$ $h^{-1}$. The ethylene yield was measured by a gas chromatograph (Shimadzu, GC-8A, Kyoto, Japan) and was expressed as $\mu L$ $C_2H_4 \cdot Kg^{-1} \cdot h^{-1}$.

### 2.2. Changes of Temperatures in Precooling

The capacity of the small VC system (Tricom International Corporation, Taipei, Taiwan) was around 120 L and could decrease the pressures to 12–14 mmHg within 20 min. RC was carried out in a walk-in refrigerator where the temperature was set to 1 °C. Before and after precooling by VC or RC, temperature distribution of the section of cabbages was measured by an infrared thermal imager (Thermography Cameras, FLIR E75, Teledyne FLIR, Wilsonville, OR, USA). The thermocouple temperature recorders (MX2303, HOBO, United States) were inserted into the centers of cabbages to measure the temperature during the precooling process.

### 2.3. Characteristics of Cabbages

After VC, each cabbage was packaged with a 0.03 mm LDPE (low-density polyethylene) bag (47 cm × 33 cm), the air was removed by a vacuum machine, and sealed as

passive MAP. The total 40 cabbages were treated as passive MAP and stored at 1 °C for 4–12 weeks. After storing for 4, 8, 10, and 12 weeks, 10 cabbages were sampled for each survey. A 10 mL syringe sucked the sample from the bags to measure the $O_2$ by an infrared $O_2$ analyzer (LF-200, Toray, Tokyo, Japan). A 1 mL syringe sucked the sample from the PE bags to measure the $CO_2$ content by the infrared $CO_2$ analyzer (X-stream enhanced XEGK, Rosemount, United States) and the ethylene concentration by a gas chromatograph (Shimadzu, GC-8A, Kyoto, Japan).

After harvest, samples were weighed (W1) and recorded by a digital balance (PB3002, Mettler, Columbus, OH, USA). After precooling by VC or RC, the cabbages were weighed again (W2) and the following formula was used to calculate the weight loss rate in precooling (%) = (W1 − W2) 100%/W1. After being stored for 4, 8, 10, and 12 weeks, the 5 cabbages were measured as (W3), and they were weighed (W4) after 5 shelf days. The following formula was used to calculate the weight loss rate during storage (%) = (W2 − W3) × 100%/W3. The total weight loss rate = (W1 − W4) × 100%/W1.

After VC, each cabbage was packaged with a 0.03 mm LDPE bag and sealed as passive MAP. For the control group, each bag with one cabbage was twist wrapped (slightly open). After storing at 1 °C for 4, 8, 10, and 12 weeks, 5 cabbages from the control group and 5 treated cabbages were removed to 10 °C for survey. The characteristics of the 'K-Y cross' cabbages were surveyed according to a yellowing index (0 = green, 1= leaf margin yellow, 2 = more green than yellow, 3 = yellow green, 4 = more yellow than green, and 5 = yellow), an inner decay index (0 = no symptom, 1 = 1–20%, 2 = 21–40%, 3 = 41–60%, 4 = 61–80%, and 5 = 81–100% of inner leaves area), a pepper spot index (0 = no symptom, 1 = 1–20%, 2 = 21–40%, 3 = 41–60%, 4 = 61–80%, and 5 = 81–100% of outer and inner head area), grey rib index (0 = no symptom, 1 = 1–20%, 2 = 21–40%, 3 = 41–60%, 4 = 61–80%, and 5 = 81–100% of inner leaves area), and flavor score (1-very bad, 2-bad, 3-medium, 4-good, and 5-very good). The leaves with yellowing, pepper spots, or shrinkage were trimmed and their weight and number recorded after removing to 10 °C for 5 days. A colorimeter (MiniScan XR Plus, 4500S, Reston, VA, USA) was used to measure the L, a*, and b* values and hue angle. One top site and two side points of each cabbage were measured and averaged as one replication to a total of 5 replications per treatment. The brightness (L) ranges from 0 to 100, and the brighter sample has a higher score in brightness. On the horizontal axis, positive a* indicates a hue of red-purple and negative a* of bluish-green. On the vertical axis, positive b* indicates yellow and negative b* blue [14]. The hue angle of 0° indicates red, 90° indicates yellow, 180° indicates green, and 270° indicates blue.

### 2.4. Statistical Analysis

The experimental data were analyzed by CoStat software 6.45 version (CoHort Software, Birmingham, UK) and subjected to the ANOVA model. Mean values were compared by least significant difference (LSD) tests at the 5% ($p < 0.05$) level of significance. Mean values of cabbage characteristics between control groups and MAP were compared by *t*-test.

### 3. Results

#### 3.1. Respiration Rate and Ethylene Production at Different Temperatures

The respiration rate and ethylene yield increased with temperature. The highest respiration rate was at 30 °C and decreased with the temperatures until 10 °C (Table 1). Cabbages showed the lowest rate of respiration between 1 and 10 °C where the respiration rates of the cabbages had no significant differences (Table 1). The respiration rate was about 4–5 times higher at 30 °C than that between 1 and 10 °C (Table 1). Ethylene production was higher between 20 and 25 °C than between 5 and 10 °C, and ethylene yield was almost double when the temperature increased from 10 °C to 20 °C (Table 1). The cabbage showed both lower respiration rate and ethylene production between 1 °C and 10 °C so that the range of temperature for the end of precooling was acceptable.

**Table 1.** The respiration rates and ethylene production of 'K-Y cross' cabbages at different temperatures in April 2022.

| Temperature (°C) | Respiration Rate (mL $CO_2$·$Kg^{-1}$·$h^{-1}$) | | Ethylene Production (µL $C_2H_4$·$Kg^{-1}$·$h^{-1}$) | |
|---|---|---|---|---|
| 1 | 11.39 ± 0.33 | e[z] | 0.46 ± 0.11 | bc |
| 5 | 10.82 ± 2.23 | e | 0.31 ± 0.11 | c |
| 10 | 14.50 ± 2.49 | e | 0.43 ± 0.07 | c |
| 15 | 26.65 ± 3.01 | d | 0.67 ± 0.26 | bc |
| 20 | 36.20 ± 2.36 | c | 1.09 ± 0.33 | a |
| 25 | 44.19 ± 1.59 | b | 0.82 ± 0.10 | ab |
| 30 | 59.16 ± 2.00 | a | 0.66 ± 0.04 | bc |

[z] Means ± SD within columns followed by the same letters are not significantly different at $p < 0.05$ by LSD test (*n* = 3).

### 3.2. Precooling Time, Thermal Images, and Weight Loss Rate

The temperature of cabbages was around 27 °C when harvested in the field. The field heat of cabbages was rapidly removed by VC, and the 1/2 precooling time and 3/4 precooling time were around 22 min and 30 min, respectively (Figure 1A). For RC, the 1/2 precooling time and 3/4 precooling time were around 340 min and 600 min, respectively (Figure 1B). Because the stable and low respiration rate was located between 1 °C and 10 °C, the 7 °C (3/4 precooling times) was set as the end of precooling process when the cooling rate of VC was about 19 times faster than that of RC. Thermal views of cabbage sections before and after cooling to around 7 °C (3/4 precooling time) by both VC and RC are recorded in Figure 2. The temperature distribution from surface to central parts after VC (Figure 2B) showed as more uniform than that by RC (Figure 2C). The differences of temperatures between surface and central parts were within 5 °C by VC but within 6.4 °C by RC at 3/4 precooling times (Figure 2B,C). After precooling, the central part of cabbages had higher temperature than the surface because heat exchange started in the outer leaves. However, a comparison of weight loss rate was made between VC and RC at 3/4 precooling time, and the former lost more weight than the latter (Table 2).

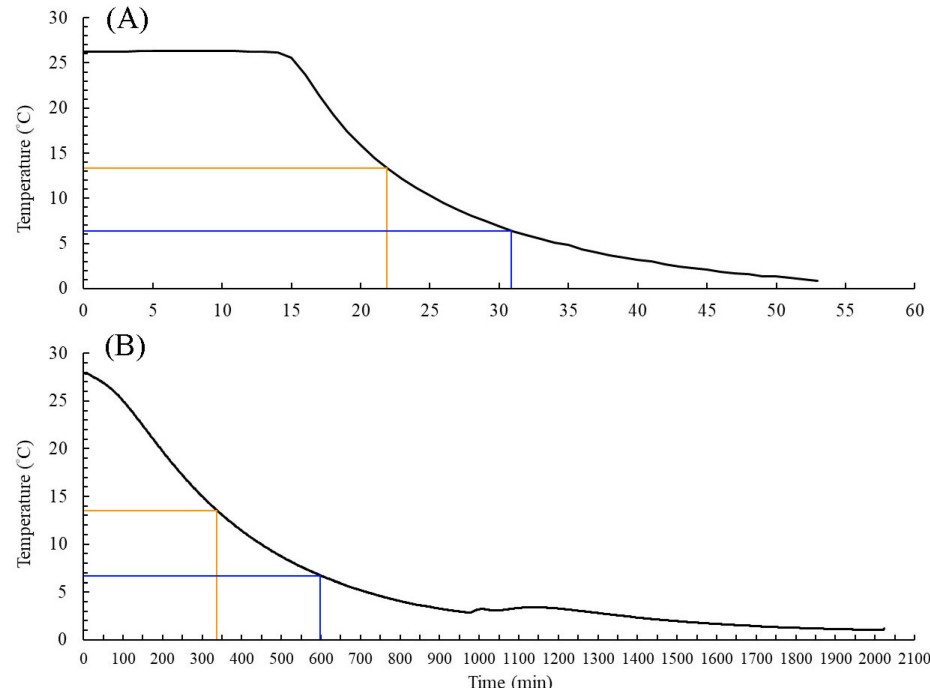

**Figure 1.** The precooling of the cabbages by vacuum precooling (**A**) and room precooling (**B**) groups. The orange and blue lines indicated the 1/2 and 3/4 precooling times, respectively.

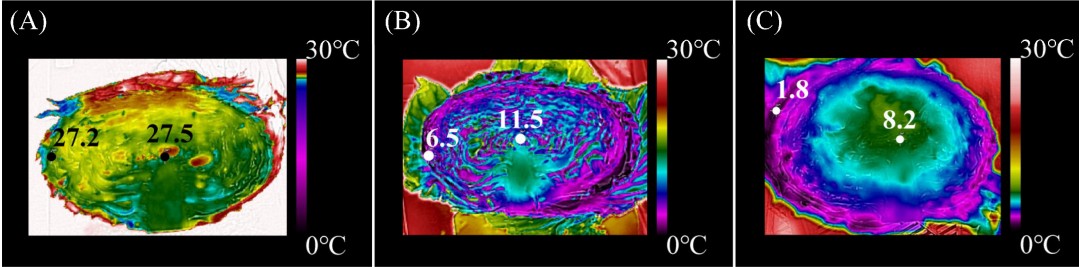

**Figure 2.** Thermal images of cabbage sections at harvest (**A**) and the temperature distribution after vacuum precooling (**B**) was shown to be more uniform than that by room precooling (**C**) at 3/4 precooling times. The differences of temperatures from surface to central parts were within 5 °C by vacuum precooling (**B**) but within 6.4 °C by room precooling (**C**).

**Table 2.** The weight loss rate during precooling and precooling times of room precooling (RC) and vacuum precooling (VC).

| Treatment | Weight Loss Rate in Precooling (%) | Time (Min) |
|:---:|:---:|:---:|
| VC | 3.48 | 31 |
| RC | 0.17 | 598 |
| Sig. | *** | *** |

\*\*\*: Significant difference at $p < 0.001$ by *t*-test ($n = 5$).

### 3.3. MAP for Storage Characteristics

After VC, the passive MAP was applied to maintain cabbages for 4–12 weeks at 1 °C. The passive MAP showed 3.43% $CO_2$ and 4.74% $O_2$ at 1 °C after 4-week storage (Table 3). Compared with the changes of atmosphere in passive MAP at 1 °C for 4–12 weeks, the $O_2$ level after 4-week storage was twice as high as that during 8–12 weeks. The $CO_2$ and $O_2$ concentration in passive MAP became balanced during 8–12 weeks (Table 3). However, there was no significant difference in ethylene in the bags between 4 and 12 weeks (Table 3). The weight loss of passive MAP cabbages was significantly lower than that of the control groups during the whole period (Table 4). After shelving at 10 °C for 5 days, the total weight loss (storage and shelf) of cabbages with passive MAP was significantly lower than that in the control group (Table 4).

The yellowing index of the outer leaves was significantly higher in the control group than MAP when cabbages were stored at 1 °C for 10 and 12 weeks and shelved at 10 °C for 5 days (Table 5 and Figure 3). There were significant differences in color parameters between treatment and control groups at 12 weeks but not at 4–10 weeks (Table 6). The hue angle of cabbages treated with MAP decreased slower than that of the control for 12-week storage (Table 6). Similarly, the a* value showed a significant difference between treatment and control groups after 12-week storage (Table 6). For the physiological disorder, there was no significant difference in the grey ribs index between the control group and treatment. However, the pepper spot index was much higher in the control groups than in the MAP for 12-week storage (Table 5). Due to the yellowing and physiological disorder, the trimmed loss increased from 0.52 to 7.32% in MAP cabbages, but it was raised from 2.27 to 9.92% in control groups after storage for 8–12 weeks and exhibition at 10 °C for 5 days (Table 4). Therefore, cabbages treated with MAP significantly preserved the quality characteristics and maintained weight, especially for 10–12 weeks.

**Table 3.** The $O_2$ and $CO_2$ concentration in passive MAP during storage at 1 °C for 4–12 weeks.

| Storage Time | $O_2$ (%) | $CO_2$ (%) | $C_2H_4$ (ppm) |
|:---:|:---:|:---:|:---:|
| 4 W | 4.74 ± 0.87 a [z] | 3.43 ± 0.62 a | 0.50 ± 0.23 a |
| 8 W | 2.61 ± 0.53 b | 4.42 ± 0.62 a | 0.47 ± 0.37 a |
| 10 W | 2.01 ± 0.46 b | 4.65 ± 0.71 a | 0.58 ± 0.17 a |
| 12 W | 1.89 ± 0.24 b | 4.05 ± 0.58 a | 0.43 ± 0.09 a |

[z] Means ± SD within columns followed by the same letter are not significantly different at $p < 0.05$ by LSD test ($n = 10$).

**Table 4.** The weight loss in storage, total weight loss, trim loss, and trimmed leaves of 'K-Y cross' cabbages stored at 1 °C from 4 to 12 weeks and shelved at 10 °C for 5 days.

| Storage Time | Treatment | Weight Loss in Storage (%) | Total Weight Loss (%) | Trim Loss (%) | Trimmed Leaf No. |
|---|---|---|---|---|---|
| 4 W + 5 D | Control | 0.13 | 0.26 | 0.00 | 0.00 |
| | Passive MAP [z] | 0.06 | 0.12 | 0.00 | 0.00 |
| | Sig. | ** [y] | * | NS | NS |
| 8 W + 5 D | Control | 0.17 | 2.55 | 2.27 | 1.00 |
| | Passive MAP | 0.15 | 0.72 | 0.52 | 0.20 |
| | Sig. | ** | * | * | * |
| 10 W + 5 D | Control | 0.24 | 8.14 | 7.87 | 3.10 |
| | Passive MAP | 0.19 | 4.34 | 4.10 | 1.60 |
| | Sig. | ** | * | * | * |
| 12 W + 5 D | Control | 0.33 | 10.26 | 9.92 | 3.90 |
| | Passive MAP | 0.24 | 7.56 | 7.32 | 2.90 |
| | Sig. | * | * | * | * |

[z] MAP = modified atmosphere packaging. [y] NS, *, **: Nonsignificant and significant at $p < 0.05$ and $< 0.01$, respectively ($n = 5$).

**Table 5.** Quality characteristics of 'K-Y cross' cabbages treated with passive MAP and control groups stored at 1 °C from 4 to 12 weeks and shelved at 10 °C for 5 days.

| Storage Time | Treatment | Yellowing Index [y] | Inner Decay Index [x] | Pepper Spot Index [w] | Grey Rib Index [v] | Flavor Score [u] |
|---|---|---|---|---|---|---|
| 4 W + 5 D | Control | 0.0 | 0.0 | 0.0 | 0.0 | 5.0 |
| | Passive MAP [z] | 0.0 | 0.0 | 0.0 | 0.0 | 5.0 |
| | Sig. | NS [y] | NS | NS | NS | NS |
| 8 W + 5 D | Control | 0.5 | 0.0 | 0.3 | 0.0 | 5.0 |
| | Passive MAP | 0.2 | 0.0 | 0.0 | 0.0 | 5.0 |
| | Sig. | NS | NS | NS | NS | NS |
| 10 W + 5 D | Control | 1.2 | 0.0 | 0.3 | 0.0 | 5.0 |
| | Passive MAP | 0.3 | 0.0 | 0.3 | 0.0 | 5.0 |
| | Sig. | ** | NS | NS | NS | NS |
| 12 W + 5 D | Control | 1.6 | 0.0 | 1.8 | 0.1 | 5.0 |
| | Passive MAP | 0.9 | 0.0 | 0.3 | 0.1 | 4.9 |
| | Sig. | * | NS | *** | NS | NS |

[z] MAP = modified atmosphere packaging. NS, *, **, ***: Nonsignificant and significant at $p < 0.05$ and $< 0.01$, respectively. ($n = 5$). [y] Yellowing index: 0 = green, 1 = leaf margin yellow, 2 = more green than yellow, 3 = yellow green, 4 = more yellow than green, 5 = yellow. [x] Inner decay index: 0 = no symptom, 1 = 1–20%, 2 = 21–40%, 3 = 41–60%, 4 = 61–80%, 5 = 81–100% of inner leaves area. [w] Pepper spot index: 0 = no symptom, 1 = 1–20%, 2 = 21–40%, 3 = 41–60%, 4 = 61–80%, 5 = 81–100% of outer and inner head area. [v] Grey rib index: 0 = no symptom, 1 = 1–20%, 2 = 21–40%, 3 = 41–60%, 4 = 61–80%, 5 = 81–100% of inner leaves area. [u] Flavor score: 1—very bad, 2—bad, 3—medium, 4—good, 5—very good.

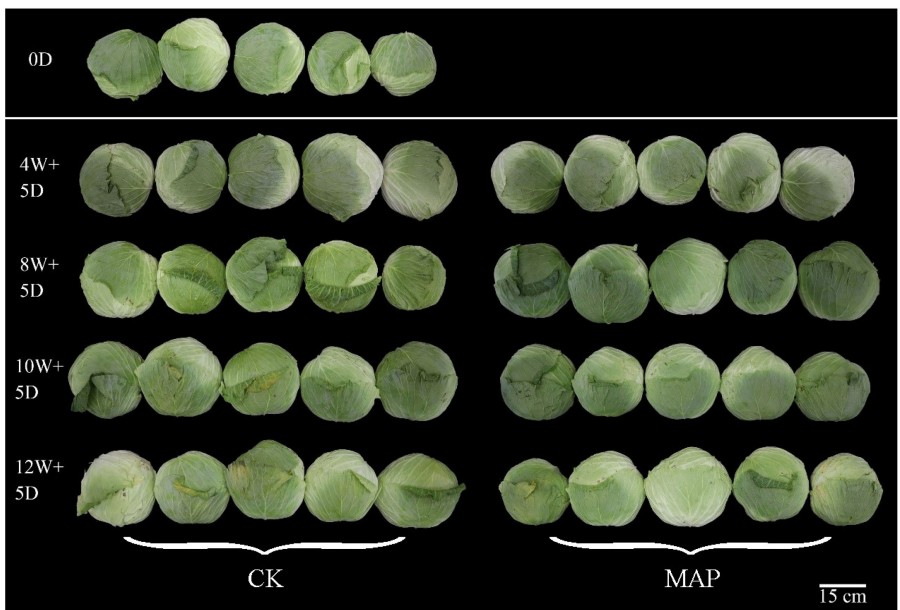

**Figure 3.** 'K-Y cross' cabbage head with passive MAP (modified atmosphere packaging) stored at 1 °C for 8–12 weeks and shelved at 10 °C for 5 days.

**Table 6.** Effects of passive MAP and control groups on changes in 'K-Y cross' cabbage color after the different storage periods and being shelved at 10 °C for 5 days.

| Storage Time | Treatment | L* | a* | b* | h° |
|---|---|---|---|---|---|
| 0 D | | 63.68 | −8.36 | 28.25 | 106.44 |
| 4 W + 5 D | Control | 69.75 | −7.43 | 31.44 | 103.33 |
| | Passive MAP z | 74.95 | −6.59 | 28.88 | 102.58 |
| | Sig. | NS | NS | * | NS |
| 8 W + 5 D | Control | 69.41 | −7.79 | 32.34 | 103.73 |
| | Passive MAP | 67.32 | −8.05 | 30.48 | 104.93 |
| | Sig. | NS | NS | NS | NS |
| 10 W + 5 D | Control | 88.87 | −3.84 | 26.09 | 97.93 |
| | Passive MAP | 79.27 | −6.44 | 27.78 | 102.71 |
| | Sig. | *** | NS | * | NS |
| 12 W + 5 D | Control | 87.12 | −2.06 | 22.19 | 95.08 |
| | Passive MAP | 83.22 | −4.02 | 25.69 | 99.04 |
| | Sig. | ** | *** | * | ** |

L* represents the lightness of a color, with 0 being black and 100 being white. a* represents the green-red axis, with negative values being green and positive values being red. b* represents the blue-yellow axis, with negative values toward blue and positive values toward yellow. h°(hue angle) corresponds to the angle between the a* and b* axes and serves as the color coordinate. [z] MAP = modified atmosphere packaging. [y] NS, *, **, ***: Nonsignificant and significant at $p < 0.05$, < 0.01, and < 0.001, respectively ($n = 5$).

## 4. Discussion

According to the respiration rate and ethylene production of 'K-Y cross' cabbages, the reasonable temperature for the end of precooling should drop below 10 °C (Table 1). Using the 3/4 precooling times as the base, the temperature of cabbages could decrease from around 27 °C to 7 °C in this study. Compared with the two precooling systems, VC was about 19 times faster than RC (Table 2), which indicated the RC needed about 10 h to arrive at the end of the precooling temperature. During this period, the high temperature caused a higher respiration rate and ethylene production, which not only consumed the carbohydrates but also lowered the quality traits such as yellowing [15]. Lettuces treated



by VC and stored at 0 °C had a shelf life of 40 days, which was almost double that by conventional methods [16]. The VC is an efficient and suitable method for the cooling of lettuces because the temperature drops of lettuce sections from the surface to the center were very similar [4]. However, the cabbages had waxy leaves and a more compact structure than lettuces. The differences of temperatures from the surface to the central parts of cabbages by VC were lower than that by VC at 3/4 precooling times (Figure 2B) and the thermal distribution in cabbages by VC was more uniform than that by RC (Figure 2C) because the VC removed heat by evaporation from all tissue, but the RC exchanged heat by conductance by cold air [15]. In this study, the VC showed the potential to precool cabbages fast and uniformly, even if the cabbages had waxy leaves and compact structure.

Although VC was more efficient for precooling cabbages than RC, the weight loss rate was higher by VC than by RC (Table 2). Because the VC depended on evaporation of water from cabbages, it was hard to avoid the weight loss of cabbages. Although this condition caused weight loss of 3.48% of cabbages, the quality of cabbages was still acceptable. Similarly, the 3.25% weight loss of baby cos lettuces still maintained acceptance and satisfaction regarding produce quality after VC [17]. Alternatively, spraying water onto the produce before the VC process can reduce the water loss [18]. For maintaining the quality characteristics, lack of weight was an acceptable trade-off for 'K-Y cross' cabbages by VC because of relative low respiration at 3/4 precooling times.

In long-term storage, vegetable quality decreases because of postharvest weight loss, yellowing, and physiological disorder. Especially, the water loss in vegetables causes shriveling and unfavorite appearance, which reduces the market value. To reduce these losses, proper packages to maintain moisture are essential in storage. The passive MAP showed a significantly lower weight loss rate than control in both storage and exhibition period because passive MAP could maintain higher moisture than the control group (Table 4). Similarly, the water loss rate of Chinese white cabbages in passive MAP was only 10%, but it reached around 45% in the control group after 5-day storage at room temperature [19]. The changes of atmosphere consisting of low $O_2$ and high $CO_2$ cause not only low respiration but also less ethylene sensitivity [20]. Ethylene promotes senescence and shortens the postharvest life even at low levels [21]. The yellowing index of outer leaves in control groups was more serious than in MAP (Table 5 and Figure 3), and this symptom was objectively supported by the a* values, which were higher in control groups after 12-week storage (Table 6). The Chinese flowering cabbage with MAP treatment had a longer shelf life than control groups because of senescence-delaying effects [22]. To avoid yellowing, the beneficial levels of 2–3% $O_2$ and 3–6% $CO_2$ for cabbages were recommended to delay chlorophyll breakdown [23]. Recently, the levels of 2% $O_2$ and 5% $CO_2$ could not only delay yellowing but also reduce physiological disorders [8]. In the present study, the passive MAP with around 2% $O_2$ and 4.5% $CO_2$ (Table 4) may have the similar function of extending the storage ability. Because the high levels of $CO_2$ and low levels of $O_2$ alleviated the pepper spots [24], this disorder formation may involve oxidative reactions. The MAP-treated Chinese flowering cabbage showed higher activity of peroxidase, superoxide dismutase, and catalase in leaves than control groups [22]. Therefore, MAP could maintain reactive oxygen species (ROS) homeostasis to mitigate oxidative damage [22]. For the physiological disorders, passive MAP alleviated the symptoms of pepper spots but not grey ribs (Table 5). Pepper spot is a disorder that is apparent on the outer leaves of the cabbages [25]. The factors involved in the development are still unclear, but its severity depends on a combination of genetic (different cultivars) and environmental factors such as high temperature and excess nitrogen fertilizer [8,10,25,26].

Due to the low cost and easy operation, MAP has been applied to extend the shelf life and sensory quality of small berry fruit [27]. To improve the shelf life of vegetables, related studies indicated that MAP was an effective method to modify the physiology and prolong shelf-life in cauliflowers [28], lettuces [13], and asparagus [29]. In contrast with passive MAP, the active MAP can achieve the suitable equilibrium atmospheres by flushing the desired gas mixtures inside packages. The shelf life of pomegranate arils at

5 °C was 12 days by active MAP and 9 days by passive MAP [30]. Using the active MAP at an ambient temperature prolonged the shelf-life of pomegranate arils for 5 days [31]. The fresh peanut kernels packaged with 90% nitrogen and 10% oxygen (active MAP) could store at 4 °C for 120 days [32]. After treating the active MAP (40% $O_2$ and 20% $CO_2$), the shelf life of apricots was doubled and the qualitative attributes preserved at 2 °C during 28-day storage [33]. In this study, the $O_2$ concentration in passive MAP for 4-week storage was almost double that for 8-week storage (Table 3). Further studies could be conducted with active MAP to understand the difference between active MAP and passive MAP. To sum up, our results indicated that combing VC and MAP for 'K-Y cross' cabbages could maintain their quality and extend their shelf life for 12 weeks, which was almost twice the storage time of commercial storage [1].

## 5. Conclusions

Two different precooling systems have been tested, and the results showed that the VC was more efficient and uniform for removing the field heat from cabbages compared to RC. The thermal images showed that the temperature distribution of cabbages by VC was more uniform than that by RC. Although VC caused higher weight loss in cabbages than RC, it could maintain better market quality by removing heat rapidly and decreasing the respiration rate within 30 min. During storage periods, the passive MAP maintained a better quality of cabbages than in control packages by decreasing weight loss, delaying yellowing, and alleviating physiological disorders for 12 weeks. This study indicated that combing vacuum cooling and passive MAP methods for 'K-Y cross' cabbages (early variety) could extend the storage period to 12 weeks.

**Author Contributions:** Conceptualization, C.-L.C. and H.-L.L.; formal analysis, H.-L.L.; funding acquisition, C.-L.C.; methodology, J.-Y.L.; resources, W.-L.C. and M.-C.H.; software, C.-W.L.; validation, C.-W.L., W.-L.C. and M.-C.H.; visualization, H.-L.L.; writing—original draft, J.-Y.L.; writing—review and editing, C.-L.C. All authors have read and agreed to the published version of the manuscript.

**Funding:** This research was funded by the Ministry of Agriculture, grant numbers 110AS-1.3.2-ST-a6, 110AS-4.2.4-FD-Z2, and 109AS-12.4.4-FD-Z1(1).

**Data Availability Statement:** All data are presented in the manuscript.

**Acknowledgments:** The authors gratefully acknowledge expert advice from Ching-Chang Shiesh in Department of Horticulture, National Chung Hsing University and Ming-Hsien Hsieh of Tainan District Agricultural Research and Extension Station.

**Conflicts of Interest:** The authors declare no conflict of interest.

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
