# Peer review of "Extending Shelf Life of Cabbage (Brassica oleracea cv. K-Y Cross) by Using Vacuum Precooling and Modified Atmosphere Packaging"

_horticulturae, doi:10.3390/horticulturae9101096_

Round 1
Reviewer 1 Report
Dear author,
Thanks for your good work. My comments attached for improvement of the paper.
Regards

It is OK.
Author Response
Thanks for your suggestion.
The most mistakes were revised, please see the new version.
Q1: combine A and B in Figure 1
R:We tried to combine A and B, but we can’t see the 1/2 and 3/4 precooling times well because the scale is to narrow. We decided to separate it.
Q2:Table 3 what was the O2 and CO2 concentration of packages at day one?
R: Thank you for pointing this out. Because of it was vaccum MAP at initial, we can’t measure it in the first day. air penetrated the packages After 4 weeks at 1°C, it is possible for we to measure it.
A 10 mL syringe sucked the sample from the bags to measure the O2 by the infrared O2 analyzer (LF-200, Toray, Japan). (Line 99-100)
Discussion :
Q3 Use more recent and related paper
R: We added the other 4 recent paper in discussion. Please see the new version
Please see the attachment

Reviewer 2 Report
Extending shelf life of cabbages (Brassica oleracea cv. K-Y cross) by optimizing vacuum precooling and modified atmosphere packaging.
The study evaluated the effect of vacuum cooling on the storability of 'K-Y Cross F1' white head cabbage. The quality of cabbage packed in two types of packaging was also compared, during 12-week storage period. Modified atmosphere packaging (MAP) was found to have a positive effect on delaying yellowing and leaf spot development of white cabbage..
Suggestions, questions and comments to MS
THE TITLE
1. I suggest to use the singular for cabbage "cabbage” in the title
2. A more appropriate term instead of optimization would be “using” vacuum precooling and modified atmosphere packaging. There was only one VC and one type of MAP, so no optimization was done in this study.
ABSTRACT
Line 15 – 18. In the first sentence it is written that, “respiration intensity and ethylene production were measured over a wide range of temperatures from 1-30 °C”, but in the next that “the temperature of the cabbage was lowered from 27°C”. Was some of the cabbage heated instead of cooled?
Line 21-22. Did the MAP packages actually maintain an unchanged atmospheric composition of 2% O2 and 4% CO2 throughout the storage period from the beginning for 12 weeks?
Line 23. How MAP affects the rapid removal of field heat from cabbage?
INTRODUCTION
Line 29. In my opinion, it should be „According to the growing period in Taiwan, there are two types….. There are many more types of cabbage in the world.
Line 41. Precooling can improve storage ability of cabbage, but not its quality The use of appropriate post-harvest conditions and treatments is intended to maintain good cabbage quality during storage.
Line 59. Add white in the sentence, after Chinese cabbage and before cabbage
MATERIALS AND METHODS
Line 76. in December
Line 78. Under what specific conditions was the cabbage stored overnight? 1-30°C is too general.
Line 94-95. How to understand this “The cabbages in the same bags were removed air and sealed as MAP”. Whether the air was removed from the cabbage or from the bags with cabbage?
2.3 chapter
1. Was there any perforation used in the PE film packaging, or were the packages very tight.
2. Was the cabbage from the packages tested after storage to see if it was fit to eat?
3. Was it examined after storage what the center of the head looks like? After removing the air from the packs, there was a very high risk of destroying the central part of the head due to oxygen deficiency.
4. How much cabbage was packed into one PE bag, one head or more? Did the cabbage from control group look like it was wrapped in stretch film after packing?
Paragraph 102 – 107. The type of weight loss should be specified in more detail, such as weight loss during pre-cooling and provide the formula, weight loss during cold storage and provide the formula, weight loss during shelf life and provide the formula.
Line 117. Instead “worst” and ”best” should be very bad and very good.
Line 109-110. Sentence completely not understandable. The methodology should specify how many experimental objects were compared, in how many repetitions the storage experiment was conducted, and how many heads of cabbage or kilograms of cabbage there were per repetition. In addition, it is important whether only one experiment was conducted or whether it was repeated.
Line 119-121. Exchange the order of the sentences.
Line 131. What is CK? No full name.
RESULTS
Table 2. Earlier in the text it was information what is the time of 3/4 precooling, so the table does not need to repeat it. Also, is there the weight loss rate in the table or just weight loss?
Figure 1 B. After about 950 minutes of cooling, the temperature of the cabbage increased slightly. What was this caused by?
Line 170 – 171. Sentence repeated from subchapter 2.3.
Line 171-174. Regarding the first sentence, do the results refer to the concentration of gases in the cabbage or in the bags with cabbage? The second sentence not understood.
Table 3. Regarding the title. It is completely unclear whether O2 and CO2 concentrations were measured, in cabbage heads or cabbage packages.
Table 4. The table must be readable and understandable in itself, and therefore the explanation requires what in your research means weight loss, total weight loss, trim loss ? From the description in subchapter 2.3, I understood that weight loss is loss during precooling time. So why are there differences among storage periods?
Figure 3. The figure shows cabbage after different periods of storage, why is for 8 weeks in the caption?
DISCUSSION
Remove or change the sentences with the same information, that was previously given in the Results section
CONCLUSIONS
Line 264 – 266. It seems to me, that the point is, that during storage, cabbage in MAP retained better quality than in control packages. MAP packaging reduced weight loss, delayed yellowing and alleviated physiological disorders in cabbage.
In general, MS needs to be corrected (more precisely defined and better organized) in many places, so that the reader can understand everything properly and have no doubts while reading this paper.
Author Response
Thanks for your suggestion, please see the attachment
THE TITLE
- I suggest to use the singular for cabbage "cabbage” in the title
- A more appropriate term instead of optimization would be “using” vacuum precooling and modified atmosphere packaging. There was only one VC and one type of MAP, so no optimization was done in this study.
Response : Agree.
New title: Extending Shelf Life of Cabbage (Brassica oleracea cv. K-Y cross) by Using Vacuum Precooling and Modified Atmosphere (Line 1-2)
ABSTRACT
Line 15 – 18. In the first sentence it is written that, “respiration intensity and ethylene production were measured over a wide range of temperatures from 1-30 °C”, but in the next that “the temperature of the cabbage was lowered from 27°C”. Was some of the cabbage heated instead of cooled?
Response : Yes. The temperature of cabbages was around 27°C when harvest so some of them were heated but most cabbages were cooled.
Line 21-22. Did the MAP packages actually maintain an unchanged atmospheric composition of 2% O2 and 4% CO2 throughout the storage period from the beginning for 12 weeks?
Response : Because of passive MAP could not maintain the similar atmospheric composition, the change of atmospheric composition from 4 to 12 weeks showed in Table 3. The concentration of O2 in the beginning was 4% and it decreased to around 2% in the 12th weeks. The concentrations of CO2 were stable.
Line 23. How MAP affects the rapid removal of field heat from cabbage?
Response : MAP doesn’t affect the rapid removal of field heat from cabbage so “combing the VC and MAP could rapidly remove field heat and maintain the storage quality of cabbages over 3 months” was modified to “ the quality of cabbages could maintain over 3 months by combing VC and passive MAP” (Line 23).
INTRODUCTION
Line 29. In my opinion, it should be „According to the growing period in Taiwan, there are two types….. There are many more types of cabbage in the world.
Response : According to the growing period, was revised to “According to the growing period in Taiwan” (Line 29).
Line 41. Precooling can improve storage ability of cabbage, but not its quality The use of appropriate post-harvest conditions and treatments is intended to maintain good cabbage quality during storage.
Response : shelf life was revised to “storage ability” (Line 41)
Line 59. Add white in the sentence, after Chinese cabbage and before cabbage
Response : We have done in Chinese white cabbages and white cabbages (Line 61-62)
MATERIALS AND METHODS
Line 76. In December
Response : in December 2022.(Line 78)
Line 78. Under what specific conditions was the cabbage stored overnight? 1-30°C is too general.
Response : the cabbages were stored in incubators or coolers with temperatures from 1 to 30 °C (Line 80-81)
Line 94-95. How to understand this “The cabbages in the same bags were removed air and sealed as MAP”. Whether the air was removed from the cabbage or from the bags with cabbage?
Response : The cabbages in the same bags were removed air and sealed as MAP was modified to “each cabbage was packaged with a 0.03 mm LDPE (low density polyethylene) bag (47 cm × 33 cm), removed air by vacuum machine, and sealed as passive MAP.” (Line 96-97)
2.3 chapter
1.Was there any perforation used in the PE film packaging, or were the packages very tight.
Response : There was no perforation in package. The control is slightly open
2.Was the cabbage from the packages tested after storage to see if it was fit to eat?
Response : Thank you for pointing this out. Yes, it is still editable without smelly order (Table 5)
3.Was it examined after storage what the center of the head looks like? After removing the air from the packs, there was a very high risk of destroying the central part of the head due to oxygen deficiency.
Response : Yes, the center of the head looked good when the cabbages were cut.
4.How much cabbage was packed into one PE bag, one head or more? Did the cabbage from control group look like it was wrapped in stretch film after packing?
Response : We added the sentence “each cabbage was packaged with a 0.03 mm LDPE (low density polyethylene) bag (47 cm × 33 cm), removed air by vacuum machine, and sealed as passive MAP.” (Line 96-97)
Paragraph 102 – 107. The type of weight loss should be specified in more detail, such as weight loss during pre-cooling and provide the formula,weight loss during cold storage and provide the formula, weight loss during shelf life and provide the formula.
Response : Thank you for pointing this out. We rewrote this in materials and methods: After harvest, samples were weighed (W1) and recorded by a digital balance (PB3002, Mettler). After precooling by VC or RC, the cabbages were weighed again (W2) and the following formula was used to calculate the weight loss rate in precooling (%) = (W1- W2) 100%/W1. After stored for 4, 8, 10, and 12 weeks, the 5 cabbages were measured as (W3), and they were weighted (W4) after 5-day shelf. The following formula was used to calculate the weight loss rate during storage (%) = (W2- W3) × 100%/W3. The total weight loss rate = (W1- W4) × 100%/W1. (Line 104-110)
Line 117. Instead “worst” and ”best” should be very bad and very good.
Response : “worst” and ”best” was modified to 1-very bad, 2-bad, 3-medium, 4-good, and 5-very good (Line 121)
Line 109-110.Sentence completely not understandable. The methodology should specify how many experimental objects were compared, in how many repetitions the storage experiment was conducted, and how many heads of cabbage or kilograms of cabbage there were per repetition. In addition, it is important whether only one experiment was conducted or whether it was repeated.
Response : Thank you for pointing this out. We rewrote “After VC, each cabbage was packaged with a 0.03 mm LDPE bag and sealed as passive MAP. For the control group, each bag with one cabbage was twist wrapped. After storing at 1 °C for 4, 8, 10, and 12 weeks, 5 cabbages from control group and 5 treated cabbages were removed to 10 °C for survey.” (Line 111-114)
Line 119-121. Exchange the order of the sentences.
Response : A colorimeter (MiniScan XR Plus, 4500S, United States) was used to measure the L, a*, and b* values and hue angle. One top site and two side points of each cabbage were measured and averaged as one replication to a total of 5 replications per treatment. (Line 123-126)
Line 131. What is CK? No full name.
Response : CK was revised to control groups (Line 135)
RESULTS
Table 2. Earlier in the text it was information what is the time of 3/4 precooling, so the table does not need to repeat it. Also, is there the weight loss rate in the table or just weight loss?
Response : We fixed it “Table 2. The weight loss rate during precooling and precooling times of room precooling (RC) and vacuum precooling (VC).” (Line 165-166)
Figure 1 B. After about 950 minutes of cooling, the temperature of the cabbage increased slightly. What was this caused by?
Response : Defrosting may increase the temperature
Line 170 – 171. Sentence repeated from subchapter 2.3.
Response : We deleted it.
Line 171-174.Regarding the first sentence, do the results refer to the concentration of gases in the cabbage or in the bags with cabbage? The second sentence not understood.
Response : We rewrote that “After VC, the passive MAP was applied to maintain cabbages for 4-12 weeks at 1 °C. The passive MAP showed the 3.43% CO2 and 4.74% O2 at 1 °C after 4-week storage (Table 3).
Table 3. Regarding the title. It is completely unclear whether O2 and CO2 concentrations were measured, in cabbage heads or cabbage packages.
Response : We rewrote that “Table 3. The O2 and CO2 concentration in passive MAP during storage at 1 °C for 4-12 weeks.” (Line 184)
Table 4. The table must be readable and understandable in itself, and therefore the explanation requires what in your research means weight loss, total weight loss, trim loss ? From the description in subchapter 2.3, I understood that weight loss is loss during precooling time. So why are there differences among storage periods?
Response : Thank you for pointing this out. We rewrote this in materials and methods: After harvest, samples were weighed (W1) and recorded by a digital balance (PB3002, Mettler). After precooling by VC or RC, the cabbages were weighed again (W2) and the following formula was used to calculate the weight loss rate in precooling (%) = (W1- W2) 100%/W1. After stored for 4, 8, 10, and 12 weeks, the 5 cabbages were measured as (W3), and they were weighted (W4) after 5-day shelf. The following formula was used to calculate the weight loss rate during storage (%) = (W2- W3) × 100%/W3. The total weight loss rate = (W1- W4) × 100%/W1. (Line 104-110) and Table 4. The weight loss in storage, total weight loss, trim loss and trimmed leaves of ‘K-Y cross’ cabbages storaged at 1 °C from 4 to 12 weeks and shelved at 10 °C for 5 days (Line 185-186)
Figure 3. The figure shows cabbage after different periods of storage, why is for 8 weeks in the caption?
Response : We fixed it. “for 8-12 weeks” (Line 202)
DISCUSSION
Remove or change the sentences with the same information, that was previously given in the Results section
Response : Thank you for pointing this out. We revised it. please see the new version.
CONCLUSIONS
Line 264 – 266. It seems to me, that the point is, that during storage, cabbage in MAP retained better quality than in control packages. MAP packaging reduced weight loss, delayed yellowing and alleviated physiological disorders in cabbage.
Response : Yes, We emphasized this point. “During storage periods, the passive MAP maintained the better quality of cabbages than in control packages because of decreasing weight loss, delaying yellowing, and alleviating physiological disorders for 12 weeks.” (Line 274-276)

Reviewer 3 Report
A very good introduction to the topic, presenting the issues that may appear for the cabbage storage. The reference to the bibliography is also appropriate to the subject, indicating the specific cooling options with their posibilities, resulting in a perfect justification of the current approach.
This study is very important because healthy food is important to people and because producing and keeping cabbage in good conditions is, beyond health, an important cultural heritage.
In chapter 2, all values of the initial condition variables are scientifically and thoroughly described. The results nicely displayed, showing all the parameter variations during the analytical process. Good explanation of the figures and tables .
In chapter 4, some indications, as conclusions of your results, would be more visible if highlighted, for instance in line 229 “spraying….” Or in 235-236 the indication of packages that maintain moisture, etc. Overall, a good study. the whole work can be in the future more extended and also the explanations which are at time good but minimal.
Author Response
In chapter 4: line 229 “spraying….We have revised to" Alternatively, spraying water onto the produce before VC process can reduce the water loss(line 232-233) "
line 235-236 the indication of packages that maintain moisture. We have modified " The passive MAP showed significant lower weight loss rate than control in both storage and exhibition period because passive MAP could maintain higher moisture than the control group (line 239-241)" to emphasize this point.

Round 2
Reviewer 1 Report
Dear author,
Thanks for the revised paper. However, it is better to cite more recent related papers, or instead of some cited old references use new related references.
Regards
It is OK.
Author Response
Dear reviewer:
Thanks for your suggestion. I added more recent references to explain the oxidative reactions. I also discussed the active MAP and passive MAP.
Please see the attachment
Kind Regards,
Chang-Lin
